# The Total Operating Characteristic from Stratified Random Sampling with an Application to Flood Mapping

**Zhen Liu [1] and Robert Gilmore Pontius Jr [2,*]**

1   Clark Labs, Clark University, Worcester, MA 01602, USA; zhenliu26@outlook.com
2   School of Geography, Clark University, Worcester, MA 01602, USA
*   Correspondence: rpontius@clarku.edu

**Abstract:** The Total Operating Characteristic (TOC) measures how the ranks of an index variable distinguish between presence and absence in a binary reference variable. Previous methods to generate the TOC required the reference data to derive from a census or a simple random sample. However, many researchers apply stratified random sampling to collect reference data because stratified random sampling is more efficient than simple random sampling for many applications. Our manuscript derives a new methodology that uses stratified random sampling to generate the TOC. An application to flood mapping illustrates how the TOC compares the abilities of three indices to diagnose water. The TOC shows visually and quantitatively each index's diagnostic ability relative to baselines. Results show that the Modified Normalized Difference Water Index has the greatest diagnostic ability, while the Normalized Difference Vegetation Index has diagnostic ability greater than the Normalized Difference Water Index at the threshold where the Diagnosed Presence equals the Abundance of water. Some researchers consider only one accuracy metric at only one threshold, whereas the TOC allows visualization of several metrics at all thresholds. The TOC gives more information and clearer interpretation compared to the popular Relative Operating Characteristic. Our software generates the TOC from a census, simple random sample, or stratified random sample. The TOC Curve Generator is free as an executable file at a website that our manuscript gives.

**Keywords:** flood; remote sensing; stratified random sampling; Total Operating Characteristic (TOC); water index

## 1. Introduction

Floods affect more people than any other natural hazard [1]. Remote Sensing is crucial for the mitigation of floods because remotely sensed images can quickly generate complete coverage of indices that distinguish between the presence and absence of water. The Normalized Difference Vegetation Index (NDVI), Normalized Difference Water Index (NDWI), and the Modified Normalized Water Index (MNDWI) have values on the continuum from −1 to 1 in a manner that can be helpful to distinguish the presence from the absence of water. Water has higher green light reflectance and lower near-infrared reflectance, therefore NDVI values closer to −1 tend to indicate water [2]. McFeeters reversed the numerator and substituted the green band for the red band in NDVI to propose the NDWI [3]. Consequently, positive NDWI values indicate water's presence while negative values indicate water's absence. Xu modified NDWI by substituting the near-infrared band with the middle-infrared band to propose MNDWI [4]. MNDWI also has the property that positive values indicate water's presence while negative values indicate water's absence. Many authors use MNDWI for flood mapping [5,6]. Readers want to know how the various continuous indices perform to distinguish between water's presence and absence. So, researchers generate a sample of observations then visually interpret each observation to produce reference data for a binary variable of water's presence or absence. Then researchers compare each continuous index to the binary reference variable.

A typical step is to apply a threshold to convert each continuous index to a binary variable of presence versus absence. Researchers then compare the threshold's binary variable to the reference binary variable, which produces a square contingency table, also known as a confusion matrix. Researchers compute a variety of metrics from the entries in the table. For example, Feyisa et al. [7] computed the kappa coefficient, omission error, and commission error to compare the performances of MNDWI, the Automated Water Extraction Index, and Maximum Likelihood classification. Munasinghe et al. [8] computed metrics to compare the performances of NDWI, MNDWI, supervised classification, and unsupervised classification. However, there are various ways to select a threshold and its selection can influence substantially each metric's result [9]. A single threshold insufficiently represents the range of each index. Any metric that derives from a single threshold ignores potentially important information concerning the various values of the index.

Pontius Jr and Si [10] created the Total Operating Characteristic (TOC), which is a quantitative technique that reveals how various thresholds of the continuous index influence the entries in the contingency table, and thus the resulting metrics. The TOC uses an algorithm similar to the Relative Operating Characteristic (ROC) but the TOC shows more information than the ROC. Both TOC and ROC examine the ranking of the index values independently of the magnitude of the index values. Authors have applied the TOC to characterize spatial patterns and to validate predictive models [11–13], particularly urban growth models [14,15]. Our literature search revealed no publications where authors have applied the TOC to indices that derive directly from remote sensing, despite TOC's natural fit for remote sensing. The TOC would have been more informative to applications that used the popular ROC. Such applications include diagnoses and predictions for medicine, business, justice, education, meteorology, engineering, public safety, machine learning, and several other fields [16,17].

Previous methods to generate the TOC required data from a census or a simple random sample. But stratified random sampling collects data more efficiently than simple random sampling. Stratified random sampling is popular in many professions because it gives results by strata and can improve the precision of unbiased estimates for the extent [18]. Those characteristics have inspired us to develop our manuscript's methods to enable the TOC to use data from stratified random sampling.

Our manuscript presents a new methodology to combine stratified random sampling with the TOC to compare various indices. The Methods section illustrates the concepts with a hypothetical example and a practical application to flood mapping, while the concepts apply to a variety of professions. The Discussion section has five subsections that interpret the TOC, explain optimal thresholds, discuss sample design, describe the software, and recommend the next steps. The Conclusions section summarizes the most important ideas that our manuscript demonstrates.

## 2. Methods

### 2.1. Illustrative Example

Table 1 gives hypothetical data for 14 observations to illustrate the concepts. The Stratum column denotes one of three strata for stratified random sampling, where each observation is selected randomly within its stratum. We number the strata according to our suspicion of water's presence where Stratum 1 is first, meaning greatest suspicion, while Stratum 3 is last, meaning least suspicion. The Size column gives the sizes of Strata 1, 2, and 3 respectively as 20, 40, and 40 square kilometers. The Weight column reports the size of each stratum divided by the number of observations in the stratum. If the reference data is derived from a census or simple random sample, then each observation has an identical weight, which is the size of the Extent divided by the number of observations. The Reference column shows a binary variable where 1 is the code for water's presence and 0 is the code for water's absence. The Index column gives the value of each observation's index, which we envision as elevation in meters. We hypothesize that observations at lower elevations have higher suspicion to have water's presence, thus Table 1 orders the

observations from first to last in ascending elevation. Each unique index value generates a unique rank. Observations that have the same index value receive the same rank, and then the subsequent observation receives the subsequent rank. The observation at 11 meters receives the first rank. Two observations at 93 meters receive the last rank.

**Table 1.** Example data where each row describes an observation from a stratified random sample where stratum 1 has two observations, stratum 2 has eight observations, and stratum 3 has four observations.

| Observation | Stratum | Size | Weight | Reference | Index | Rank |
|---|---|---|---|---|---|---|
| 1 | 1 | 20 | 10 | 1 | 11 | 1 |
| 2 | 2 | 40 | 5 | 1 | 22 | 2 |
| 3 | 1 | 20 | 10 | 0 | 31 | 3 |
| 4 | 2 | 40 | 5 | 1 | 42 | 4 |
| 5 | 2 | 40 | 5 | 1 | 52 | 5 |
| 6 | 2 | 40 | 5 | 1 | 52 | 5 |
| 7 | 2 | 40 | 5 | 0 | 52 | 5 |
| 8 | 3 | 40 | 10 | 1 | 63 | 6 |
| 9 | 2 | 40 | 5 | 0 | 72 | 7 |
| 10 | 2 | 40 | 5 | 0 | 72 | 7 |
| 11 | 2 | 40 | 5 | 0 | 72 | 7 |
| 12 | 3 | 40 | 10 | 0 | 83 | 8 |
| 13 | 3 | 40 | 10 | 0 | 93 | 9 |
| 14 | 3 | 40 | 10 | 0 | 93 | 9 |

The TOC uses each rank to make a binary diagnosis for each observation. The TOC then compares the binary diagnosis at each rank to the binary reference variable, which generates a contingency table in the format of Figure 1. Extent is the bottom-right entry, which is a constant size for any case study. Abundance is the size of presence according to the reference variable, which is also a constant size for any case study. If the reference data is derived from a sample, then Abundance is an estimate. Each rank $j$ applied to the index determines the six entries that have subscript $j$ in the contingency table. Rank $j$ diagnoses presence for observations that have ranks less than or equal to $j$. Thus, Diagnosed Presence is smaller at earlier ranks and larger at later ranks. Hits are the intersection of diagnosed presence and reference presence. Extent, Abundance, Diagnosed Presence, and Hits determine the other entries in the contingency table. False Alarms are the intersection of diagnosed presence and reference absence. False Alarms are both commissions of presence error and omission of absence error. Misses are the intersection of diagnosed absence and reference presence. Misses are both commission of absence error and omission of presence error. Correct Rejections are the intersection of diagnosed absence and reference absence. The sum of Hits and Misses is Abundance. The sum of Hits and False Alarms is Diagnosed Presence. The sum of Hits, False Alarms, Misses, and Correct Rejections is Extent. The TOC shows the nine entries in the contingency table at each threshold. Table 2 defines the mathematical notation that the subsequent equations use to construct the TOC.

| | | **Reference** | | |
|---|---|---|---|---|
| | | **Presence** | **Absence** | **Sum** |
| **Diagnosis** | **Presence** | Hits$_j$ | False Alarms$_j$ | Diagnosed Presence$_j$ |
| | **Absence** | Misses$_j$ | Correct Rejections$_j$ | Extent − Diagnosed Presence$_j$ |
| | **Sum** | Abundance | Extent − Abundance | Extent |

**Figure 1.** Contingency table of sizes at rank $j$.

**Table 2.** Mathematical notation for Equations (1)–(5).

| Symbol | Meaning |
|---|---|
| $D_j$ | Size of Diagnosed Presence at rank $j$ |
| $H_j$ | Size of Hits at rank $j$ |
| $j$ | Rank for a point on the TOC curve where 0 indicates the point at the origin, 1 indicates the first point near the origin, and $J$ indicates the last point at the top right of the TOC |
| $J$ | Number of unique index values among the observations |
| $m$ | Identifier for stratum = 1, 2, ... , $M$ |
| $M$ | Number of strata |
| $n_m$ | Number of observations in stratum $m$ |
| $O_m^j$ | Number of observations in stratum $m$ diagnosed as a presence at rank $j$ |
| $P_m^j$ | Number of observations in stratum $m$ diagnosed correctly as a presence at rank $j$ |
| $S_m$ | Size of stratum $m$ |
| $T_j$ | Threshold value at rank $j$ |
| $W_m$ | Weight as size per observation for observations in stratum $m$ |

$$W_m = S_m / n_m \tag{1}$$

$$\text{Size of Extent} = \sum_{m=1}^{M} S_m = \sum_{m=1}^{M} \left( O_m^J W_m \right) = D_J \tag{2}$$

$$\text{Size of Abundance} = \sum_{m=1}^{M} \left( P_m^J W_m \right) = H_J \tag{3}$$

$$D_j = \sum_{m=1}^{M} (O_m^j W_m) \tag{4}$$

$$H_j = \sum_{m=1}^{M} (P_m^j W_m) \tag{5}$$

Equation (1) computes the weight for each stratum's observations as the size of the stratum divided by the number of observations in the stratum, meaning each stratum's weight is the inverse of the stratum's sampling intensity. If the data were observations from a census or simple random sample, then all observations have the same weight. Equation (2) expresses the size of the Extent as the sum of the sizes of the strata, which equals the size of Diagnosed Presence at the last rank. Equation (3) estimates the size of Abundance as the size of Hits at the last rank. Equations (4) and (5) are functions of the rank. Each rank generates a contingency table, which generates a point on the TOC curve. The TOC curve is a sequence of line segments that connect point $j-1$ with point $j$ when $0 < j \leq J$.

Each rank corresponds to a threshold, which is one of the index values. If a smaller index value indicates a greater suspicion of reference presence, then $T_0 = -\infty$ and $T_{j-1} < T_j$ when $0 < j \leq J$. If a larger index value indicates a greater suspicion reference presence, then $T_0 = \infty$ and $T_{j-1} > T_j$ when $0 < j \leq J$. For the data in Table 1, $T_0 = -\infty$, $T_1 = 11$, and $T_9 = 93$. If the user lacks suspicion concerning the relationship between an index and presence, then we recommend the user choose the option where a smaller index value indicates greater suspicion of presence because the sequence of thresholds where $T_{j-1} < T_j$ is easier to envision.

For Table 1, Equation (1) computes the weight for observations in Stratum 1 as 20/2 = 10 square kilometers per observation. Also from Equation (1), the weight for observations in Stratum 2 is 40/8 = 5 square kilometers per observation and in Stratum 3 is 40/4 = 10 square kilometers per observation. Equation (2) computes the size of the Extent as 100 square kilometers. Equation (3) estimates the size of Abundance as 40 square kilometers.

Figure 2 shows the TOC space for the data in Table 1. The lengths of vertical and horizontal axes are equal for visual clarity, while the maximum value on the vertical axis is less than the maximum value on the horizontal axis. The maximum value on the horizontal axis is the size of the Extent. The horizontal axis shows the Diagnosed Presence as the sum of Hits and False Alarms, which must be between zero and the size of the Extent. The maximum value on the vertical axis is the size of Abundance. The vertical axis shows Hits, which must be between zero and the size of Abundance. Any TOC curve must reside in or on the white parallelogram of the TOC space. The sum of Hits and Misses is the height of the parallelogram, while the sum of False Alarms and Correct Rejections is the width of the parallelogram. The parallelogram's bottom boundary is where Hits equal zero, while its top boundary is where Misses equal zero. The parallelogram left boundary is where False Alarms equal zero, while its right boundary is where Correct Rejections equal zero. Gray shows regions that are impossible for a TOC curve to reside. The left gray triangle is an impossible region because Hits cannot be greater than Hits plus False Alarms. The right gray triangle is an impossible region because False Alarms plus Correct Rejections is a constant, which equals the Extent minus Abundance. The sizes of the Extent and Abundance dictate the TOC parallelogram within the square space. We usually code the reference variable as 1 for presence and 0 for absence so that absence constitutes half or more of the Extent because those codes generate a wider parallelogram than if the codes were the opposite for presence and absence. For example, if an Extent contains 90% water, then codes of 1 for non-water and 0 for water would produce a larger TOC parallelogram than codes of 1 for water and 0 for non-water. Larger TOC parallelograms are more revealing visually, while the TOC works for either coding format.

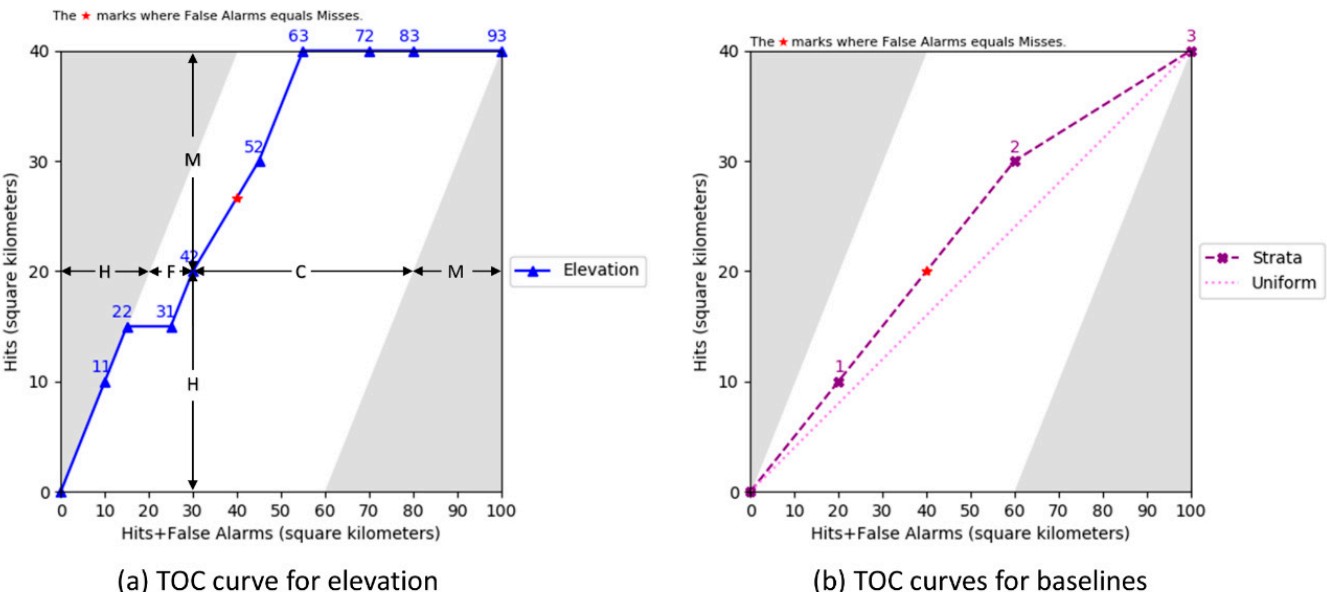

(a) TOC curve for elevation  (b) TOC curves for baselines

**Figure 2.** TOC curves for (**a**) elevation and (**b**) two baselines. The labels on the points give the threshold values. Hits + False Alarms is the Diagnosed Presence. The size of Hits is *H*, Misses is *M*, False Alarms is *F*, and Correct Rejections is *C*.

Equations (4) and (5) compute the coordinates of the points on the TOC curve. All TOC curves begin at the origin of the TOC space with the point that derives from rank 0. Subsequent ranks generate a sequence of points that progress from left to right as the Diagnosed Presence grows. The increase in the horizontal coordinate between point $j-1$ and point $j$ on the TOC curve is the sum of the weights of the observations that contribute to point $j$ but not to point $j-1$ when $0 < j \leq J$. Each point corresponds to a table that has the format of Figure 1. If the data derive from a census, then the table's entries give the population's information. If the data derive from a sample, then the table's entries estimate the population's information. Figure 2a shows the TOC curve for the example. Consider

the point ranked fourth with a threshold of 42. Four observations have elevations that are less than or equal to 42, consisting of two observations in Stratum 1 and two observations in Stratum 2. The horizontal coordinate of the point ranked fourth is $D_4 = 2 \times 10 + 2 \times 5 = 30$ square kilometers. In those four observations, the numbers of reference presence observations in Strata 1 and 2 are respectively one and two. Thus, the vertical coordinate of the point ranked fourth is Hits, which is $H_4 = 1 \times 10 + 2 \times 5 = 20$ square kilometers. The size of Hits is the vertical coordinate of a point on a TOC curve. Misses is Abundance minus Hits. The size of Misses is the vertical distance between a point and the maximum value on the vertical axis. The size of False Alarms is the horizontal distance between a point and the parallelogram's left boundary. The size of Correct Rejections is the horizontal distance between the point and the parallelogram's right boundary. A threshold of 42 in the example generates a point where the size of Hits is 20, Misses is 20, False Alarms is 10, and Correct Rejections is 50 square kilometers. The last rank generates the last point, which is always the top right corner of the parallelogram where all observations are Diagnosed Presence. Both Misses and Correct Rejections are zero at the top right corner, where the Diagnosed Presence equals the Extent and where Hits equals Abundance.

Figure 2a is helpful to understand the diagnostic ability of an index. A baseline diagnosis could enhance interpretation. Figure 2b gives two baselines, called Uniform and Strata. The Uniform baseline is a single diagonal line from the origin to the top right corner of the TOC parallelogram. The Uniform line is a TOC curve of the mathematical expectation for an index consisting of random numbers.

Figure 2b shows a second baseline that is relevant to a stratified Extent. Each stratum corresponds to a point on the TOC curve. Observations that belong to a stratum have the same rank as the stratum. Strata 1, 2, and 3 in the example have ranks in that order with 1 as first. Take Stratum 2 as an example to show how to calculate the coordinates of points on the TOC curve for Strata. The Stratum column in Table 1 gives the rank of each observation when we draw the TOC curve for strata. The rank derives from the stratum identifier rather than the elevation index. Each observation in Strata 1 or 2 has a rank less than or equal to 2. Two observations are in Stratum 1 and eight observations are in Stratum 2. The horizontal coordinate of the point ranked second is $2 \times 10 + 8 \times 5 = 60$ square kilometers. In those 10 observations, the number of reference presence observations in Strata 1 and 2 respectively are one and four. Thus, the vertical coordinate of the point ranked second is $1 \times 10 + 4 \times 5 = 30$ square kilometers.

The red stars in Figure 2 show the points on the TOC curve where Diagnosed Presence equals Abundance. Diagnosed Presence is Hits plus False Alarms while Abundance is Hits plus Misses. Therefore, False Alarms equals Misses at the red stars, which are points where quantity difference is zero while allocation difference accounts for all the difference [13]. Thresholds to the left of the red star are where Diagnosed Presence is smaller than Abundance, while thresholds to the right of the red star are where Diagnosed Presence is larger than Abundance.

The Area Under the Curve (AUC) is a metric that summarizes the strength of the overall diagnostic ability of an index's ranks. AUC is a ratio where the numerator is the area within the parallelogram that is under the TOC curve while the denominator is the area of the parallelogram. A larger AUC indicates stronger positive diagnostic ability integrated across the ranks. The AUC is always one half for a Uniform TOC. In the example, the AUC is 0.53 for Strata and 0.82 for Elevation.

### 2.2. Practical Example

A bomb cyclone caused catastrophic floods in the Midwestern United States during March 2019 [19]. Floods devastated the region near the confluence of the Missouri and Platte Rivers in the state of Nebraska, particularly in the city of Plattsmouth 41° N 96° W. We apply the Total Operating Characteristic to help to map the 2019 flood near Plattsmouth. The strata map derives from floods preceding 2019. Stratum 1 is the region that is usually water, such as rivers and ponds. Stratum 2 is the region that is usually not water but was

water during floods preceding 2019. Stratum 3 is the region that is usually not water and remained not water during floods preceding 2019. We took a stratified random sample of 200 observations where we conducted a visual assessment of Sentinel-2 images to determine each observation's binary reference variable, which is water's presence or absence on 16 March 2019. Stratum 1 has 50 observations, all of which are water according to the visual assessment. Stratum 2 has 100 observations of which 79 are water, while Stratum 3 has 50 observations of which 10 are water. The sampling intensity in each stratum is the number of observations per square kilometer of each stratum. Sampling intensity is greatest in Stratum 1 and least in stratum 3. Figure 3a shows the strata and observation points. Other parts of Figure 3 show the three indices: Normalized Difference Vegetation Index (NDVI), Normalized Difference Water Index (NDWI), and the Modified Normalized Water Index (MNDWI). Equations (6)–(8) define the three indices as functions of spectral reflectance values where $N$ denotes near-infrared, $R$ denotes red, $G$ denotes green, and $S$ denotes shortwave infrared regions [2–4].

$$\text{Normalized Difference Vegetation Index} = \frac{N - R}{N + R} \tag{6}$$

$$\text{Normalized Difference Water Index} = \frac{G - N}{G + N} \tag{7}$$

$$\text{Modified Normalized Difference Water Index} = \frac{G - S}{G + S} \tag{8}$$

The indices derive from the Sentinel-2 images that are the source of the reference data. The spatial extent of the maps in Figure 3 has 512 rows and 512 columns of pixels, where each pixel has a 30-meter resolution. The continuous legend in Figure 3b–d has blue to signify the index values ranked earlier in the diagnosis of water and black to signify index values ranked last in the sequence. In Figure 3b–d, blue indicates negative values for NDVI and positive values for both NDWI and MNDWI. Purple is an index value of zero. The TOC measures the degree to which water's presence is associated with lower values of NDVI and higher values of NDWI and MNDWI. Each of the histograms for the three indices shows the distribution of the index values and the strata from which the values derive. Smaller NDVI values derive mostly from Strata 1 and 2 while larger NDVI values derive mostly from Stratum 3. For NDWI and MNDWI, smaller index values derive mostly from Stratum 3 while larger index values derive mostly from Strata 1 or 2. The histograms show that MNDWI is better than the other indices at distinguishing stratum 3 from the other strata. Researchers consider where along the horizontal axis of the histogram to select a threshold that distinguishes between water's presence from water's absence. Water's Diagnosed Presence is to the left of a threshold for NDVI and to the right of a threshold for NDWI and MNDWI in the histograms in Figure 3. The TOC curves show how the selection of each possible threshold influences the sizes of Diagnosed Presence and Hits.

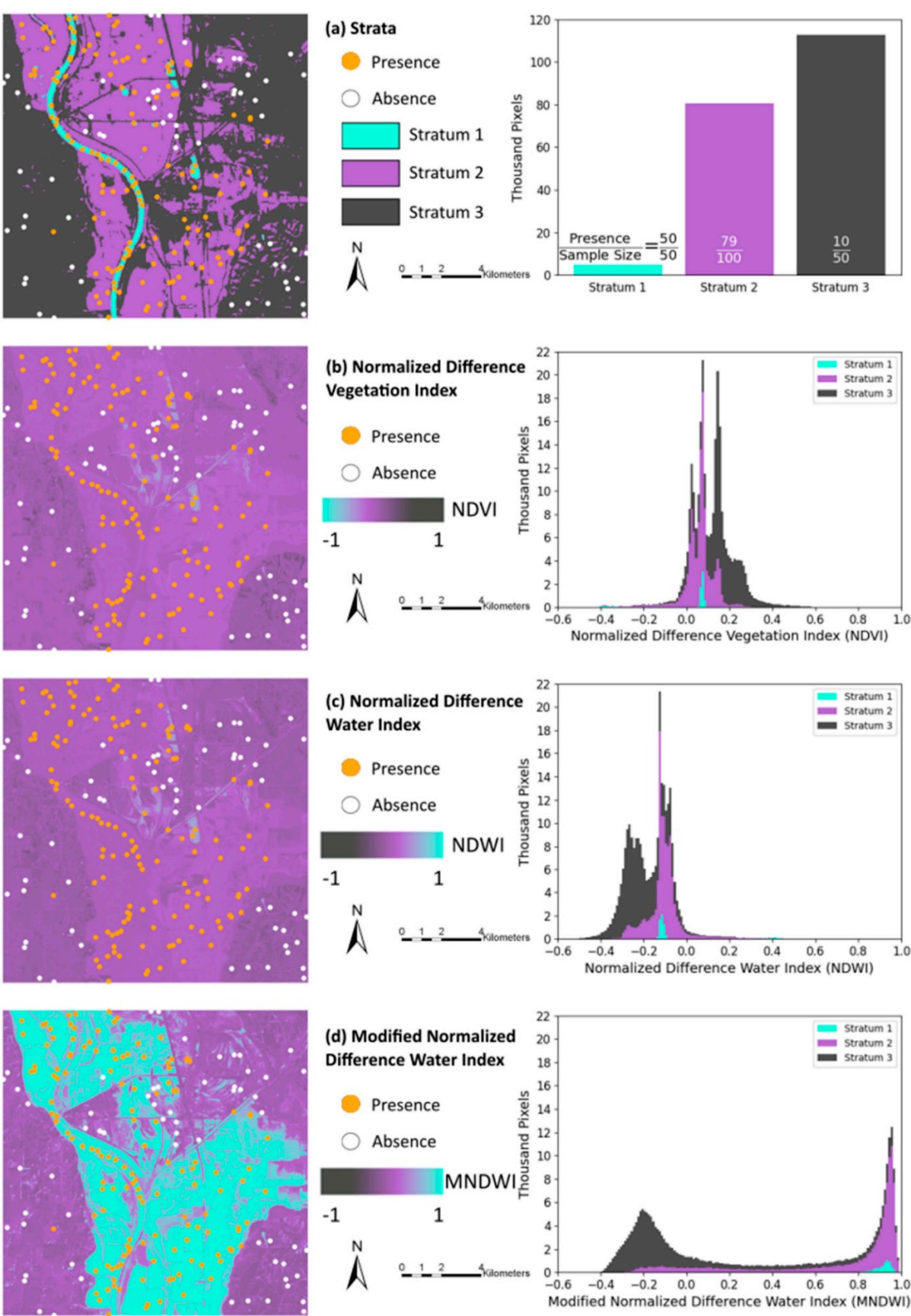

**Figure 3.** Data from (**a**) Strata, (**b**) NDVI, (**c**) NDWI, and (**d**) MNDWI.

## 3. Results

Figure 4 shows the TOC curves for two baselines and three indices. The size of the Extent is the maximum value on the horizontal axis in Figure 4a. The maximum value on the vertical axis is the size of the Abundance, which is the estimated size of the water area in the Extent according to the stratified reference data. The sizes of the Extent and Abundance are approximately 236 and 109 square kilometers respectively, which implies that water covers 46% of the Extent. Each red star is directly below the parallelogram's top left corner, which has a horizontal coordinate equal to the Abundance.

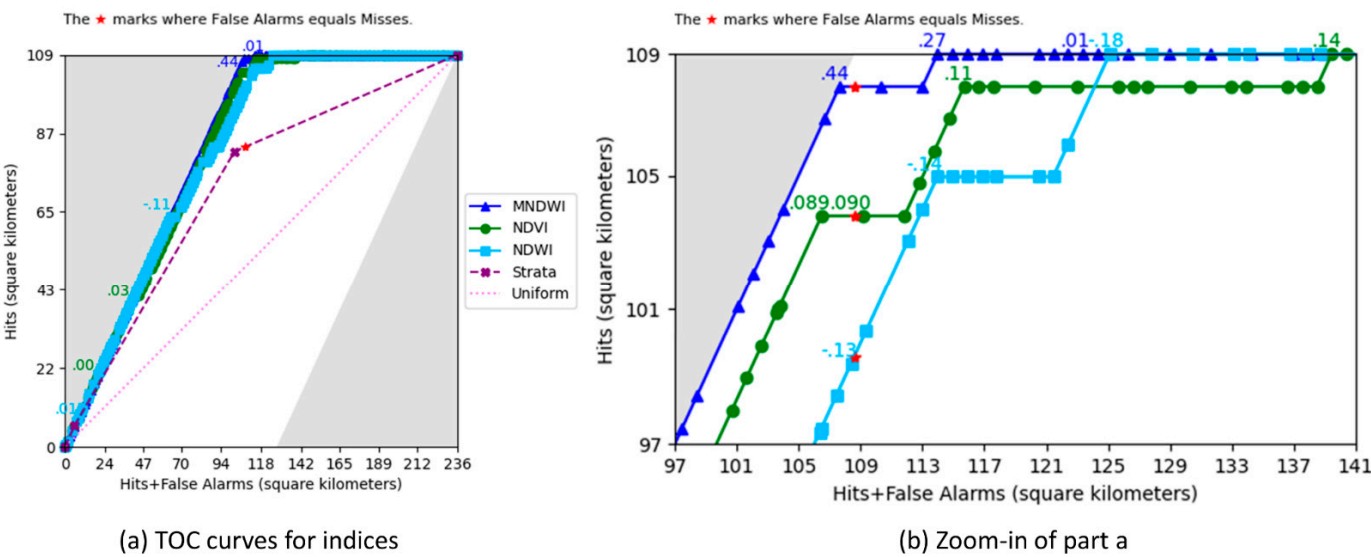

(a) TOC curves for indices    (b) Zoom-in of part a

**Figure 4.** TOC curves showing (**a**) the full TOC parallelogram and (**b**) zoom-in of the top left corner of the TOC parallelogram.

The Uniform line is a baseline that portrays the expected TOC curve for an index that has random values. The Strata TOC curve is a more relevant baseline that ranks stratum 1 first and stratum 3 last. The horizontal coordinates of the points on the Strata curve reveal the sizes of the three strata. Stratum 1 is 6 square kilometers; stratum 2 is 96 square kilometers; stratum 3 is 134 square kilometers. The union of Strata 1 and 2 was water during floods before 2019. The red star is slightly to the right of the point for stratum 2, which indicates the Abundance of water in 2019 is slightly greater than the size of previous floods.

The TOC curves for all three remotely sensed indices hug the left bound of the parallelogram because much of water's reference presence exists at the earlier ranked values of each index. The TOC curves for the indices hug the top bound of the parallelogram because most of the water's reference absence exists at the latter ranked values. Figure 4a shows labels at two thresholds for each index. One of those labeled thresholds is where the index value is closest to zero. The threshold of zero is to the left of the red star for NDWI and NDVI, which indicates the thresholds at zero for NDWI and NDVI diagnose less water than the Abundance of water. The threshold at zero is to the right of the red star for MNDWI, which indicates the threshold at zero for MNDWI diagnoses more water than the Abundance of water. The other labeled threshold on each index in Figure 4a shows where each index leaves the left bound. Those thresholds are 0.03 for NDVI, −0.11 for NDWI, and 0.44 for MNDWI. False Alarms are zero at thresholds ranked earlier than those three thresholds.

Figure 4b zooms in on the top left corner of the parallelogram. Figure 4b has a label at each index's threshold that is closest to the red star, which is where the diagnosed size of water is closest to the Abundance of water. At those thresholds, the greatest number of Hits derives from MNDWI, followed by NDVI, and NDWI. Figure 4b has a label also at

the threshold where each TOC curve arrives at the top bound, which is where the earliest rank attains zero Misses.

The TOC curve for MNDWI is at or above the TOC curves for NDWI and NDVI at all thresholds, implying that the presence of water has a stronger positive association with the rankings for MNDWI than with the rankings for NDWI or NDVI. The AUC quantifies the relationship synthesized overall thresholds. MNDWI has an AUC of 0.9996, while NDVI and NDWI follow with 0.9836 and 0.9758. The Strata baseline has an AUC of 0.8018 while the Uniform baseline has an AUC of a half.

Figure 4b uses the index values to label additional thresholds of particular interest. Sections 4.1 and 4.2 interpret the labeled thresholds.

## 4. Discussion

### 4.1. TOC Interpretation

The TOC curves show a plethora of information that allows comparison among the indices. Important information includes TOC baselines, red stars where Diagnosed Presence equals Abundance, the point where each curve leaves the parallelogram's left boundary, the point where each curve arrives at the parallelogram's top boundary, and the shape of the TOC curve.

A baseline TOC derives from a ranking that we could have made prior to knowing the indices or the reference data. The Uniform baseline portrays the expected curve for an index that has random values, which we could have used to rank each observation. The Uniform baseline uses no prior information; however, we have information concerning prior floods. The strata express information concerning prior floods, which existed before we obtained the indices or the reference data. The Strata TOC curve is a more relevant baseline than the Uniform baseline because the Strata TOC reflects the ranking based on our best available prior information. We suspected water's presence would be most intensive in stratum 1 and least intensive in stratum 3. Thus, the Strata TOC ranks stratum 1 first and stratum 3 last. The slope of the TOC segment for each stratum shows the intensity of water in each stratum because the slope of a segment is the increase in Hits divided by the increase in Diagnosed Presence from one point to a subsequent point on the TOC curve. The differences between the TOC of the Strata and the TOC curves of the remotely sensed indices reveal the value added by the indices beyond the value of the strata.

The TOC curves show a red star at the point where the Diagnosed Presence equals Abundance. Thresholds to the left of the red star diagnose less than the estimated size of water's presence. Thresholds to the right of the red star diagnose more than the estimated size of water's presence. If the user wants to diagnose as close as possible to the correct size of water for any particular index, then the user should select the threshold closest to the red star. Figure 4b shows that the area diagnosed as water is approximately 109 square kilometers at thresholds of 0.44 for MNDWI, 0.090 for NDVI, and −0.13 for NDWI. MNDWI has more Hits than the other two indices at that diagnosed size.

Additional important points on each TOC curve include the point where the curve leaves the parallelogram's left boundary and the point where the curve arrives at the parallelogram's top boundary. The size of False Alarms at any threshold is the horizontal distance between the point on the curve and the left boundary. The size of Misses at any threshold is the vertical distance between the point on the curve and the maximum value on the vertical axis. Figure 4 shows that the NDWI curve leaves the left boundary when the curve's threshold equals −0.11 and arrives at the top boundary when the threshold equals −0.18. False Alarms and Misses exist simultaneously at thresholds between −0.11 and −0.18. We suspect that some False Alarms are on built land because built land and water appear similar in the spectral bands that NDWI uses [3]. This could be the reason why NDWI has the smallest AUC among the three indices. The TOC curves show how the thresholds generate False Alarms and Misses, which allow users to understand how an index generates errors. This understanding might allow scientists to improve the formulation of an index.

### 4.2. Optimal Threshold and Summary Metrics

Some projects call for a single map of presence versus absence by applying a threshold to convert the index map to a binary map. Many possible thresholds exist, thus the project's investigators must use a criterion to select a threshold. This criterion is the de facto definition of optimal for the project. The profession should not have a universal mathematical definition for optimal because the goals of each specific project should guide the definition of optimal for each project. Various people might have differing opinions concerning a project's goals, thus the requirement to select a criterion should generate a conversation among scientists and stakeholders. The conversation to define optimal is likely to be challenging and enlightening because the conversation will force the project's team to clarify their goals. The conversation might reveal that the criterion must weigh various competing goals. Table 3 lists some metrics that would enlighten the conversation and other popular metrics that could muddy the conversation.

**Table 3.** Summary metrics to inspire a conversation to define an optimal threshold. The size of Hits is *H*, Misses is *M*, False Alarms is *F*, and Correct Rejections is *C*. The cost of each False Alarm is *f* while the cost of each Miss is *m*.

| Criterion to Select Threshold | Metric's Expression |
|---|---|
| Minimum Quantity Difference | $\lvert F - M \rvert$ |
| Minimum Weighted Cost | $f\,F + m\,M$ |
| Maximum Correct | $H + C$ |
| Maximum Odds Ratio | $\frac{HC}{FM}$ |
| Maximum Intersection over Union | $\frac{H}{M+H+F}$ |
| Maximum F1 Score | $\frac{2H}{M+2H+F}$ |
| Maximum Kappa | $\frac{2(HC-FM)}{(H+F)(F+C)+(H+M)(M+C)}$ |
| Maximum Phi | $\frac{HC-FM}{\sqrt{(H+F)(M+C)(H+M)(F+C)}}$ |

We encourage researchers to consider quantity difference, which is the first metric in Table 3. A classification has zero quantity difference when the size of Diagnosed Presence equals the size of Abundance. If the classification's only goal is to diagnose the size of Abundance, then Minimum Quantity Difference is the only criterion to select an optimal threshold. The threshold closest to the red star in Figure 4 gives the threshold at the Minimum Quantity Difference for each index. These thresholds are 0.44 for MNDWI, 0.090 for NDVI, and −0.13 for NDWI.

The second criterion in Table 3 is the Minimum Weighted Cost. If the cost of a False Alarm has the same weight as the cost of a Miss, then the threshold that minimizes the weighted cost is the threshold that minimizes the sum of False Alarms and Misses. If the cost of a False Alarm equals the cost of a Miss, then MNDWI produces the minimum cost among all indices at its threshold of 0.44 where False Alarms are 0 and Misses are 1 square kilometer. However, the cost of a False Alarm might differ from the cost of a Miss depending on the application. For applications to flood mapping, False Alarms could cause rescuers to allocate resources to places where victims are absent; while Misses could cause rescuers to neglect places where victims are present. For this application, the cost of a Miss seems greater than the cost of a False Alarm, thus a map that has more False Alarms than Misses could be better than a map that has more Misses than False Alarms. The cost of a Miss relative to the cost of a False Alarm is a matter of debate and should depend on the particular application. For example, if the application concerns the threshold that engineers set for household smoke detectors, then the cost of a False Alarm is the annoyance of loud noise, while the cost of a Miss could be death. If a Miss is twice as costly as a False Alarm, then $m/f = 2$, which generates a criterion sufficient to select an optimal threshold. It is not necessary to express the costs of both Misses and False Alarms as particular units, such as dollars.

A TOC curve is convex when a line segment between any two non-consecutive thresholds intersects the TOC curve at only those two thresholds. Figure 4 shows that the TOC curve for Strata is convex but the TOC curves for the three indices are not convex. If a TOC curve is convex and above the Uniform line, then the weighted cost attains its minimum either at one threshold or along one segment of two consecutive thresholds. NDVI's curve is not convex because a line through the thresholds 0.089 and 0.14 intersects NDVI's curve between those thresholds. Thus, NDVI could have non-consecutive thresholds that minimize the weighted cost, depending on the cost of a False Alarm relative to the cost of a Miss in the linear cost function. The possible non-consecutive thresholds are on the corners of NDVI's TOC curve at thresholds 0.089, 0.11, and 0.14. False Alarms are 3 square kilometers while Misses are 5 square kilometers at threshold 0.089. False Alarms are 8 while Misses are 1 at 0.11. False Alarms are 31 while Misses are 0 at 0.14. If the weights are such that $m/f = 5/4$, then NDVI has two optimal thresholds at 0.089 and 0.11 where each threshold has a weighted cost of $9\frac{1}{4}$ times the cost of a square kilometer of False Alarms. If the weights are such that $5/4 < m/f < 23$, then NDVI has one optimal threshold at 0.11. If the weights are such that $m/f = 23$, then NDVI has two optimal thresholds at 0.11 and 0.14 where each threshold has a weighted cost of 31 times the cost of a square kilometer of False Alarms. If $0 < f < \infty$ and $m = \infty$ then the Minimum Weighted Cost is at the earliest threshold on the parallelogram's top bound where Misses are zero. If $f = \infty$ and $0 < m < \infty$ then the optimal threshold is the last threshold on the parallelogram's left bound where False Alarms are zero.

Table 3 lists some metrics that are popular to measure accuracy but have questionable appropriateness depending on the application. All of the criteria concerning maximum agreement in Table 3 treat the cost of a False Alarm as equal to the cost of a Miss. If the cost of a False Alarm equals the cost of a Miss, then the Minimum Weighted Cost criterion is identical to the Maximum Correct criterion. Maximum Correct can seem initially attractive to some researchers due to its seemingly clear interpretation. But, the Maximum Correct criterion has some potentially undesirable properties that relate to its failure to consider quantity difference. The Maximum Correct criterion might select a threshold that has a substantial quantity difference depending on the shape of the TOC curve. For example, the Uniform TOC attains its Maximum Correct either at the origin or at the upper right corner, meaning where either Misses or Correct Rejections are zero, and potentially far from the threshold at the correctly diagnosed size.

A binary classification specifies presence in terms of quantity and allocation. In a map, quantity defines the size of presence, while allocation defines where the presence resides. Pontius Jr and Millones [20] showed how to measure quantity difference distinctly from allocation difference. The allocation difference is two times the minimum of Misses and False Alarms. The total difference is the sum of quantity difference and allocation difference. If a diagnosis of the wrong size is a more serious error than a diagnosis of the wrong allocation, then quantity difference is a more serious error than allocation difference. The Maximum Correct criterion ignores that distinction because the Correct metric measures only agreement. Figure 4b shows that the Maximum Correct criterion applied to NDVI would select the threshold at 0.089 while the threshold at 0.090 has a smaller quantity difference.

Table 3 includes metrics of agreement as a warning to readers who might see those metrics in the literature. Some of those metrics appear in the literature because of dysfunctional social habits that have infected the professional community. Some authors have told us that they report those metrics because they assume that other people expect those metrics. This perpetuates social dysfunction. This justification violates the principle that a criterion should reflect each project's specific goals. Various projects can have different goals, so it makes no scientific sense for all projects to use the same metrics. Furthermore, several popular metrics have undesirable mathematical properties. For example, the Odds Ratio is undefined when either False Alarms or Misses are zero. The last four indices in Table 3 are ratios that can range from −1 to 1, where 1 means perfect agreement. It is not

clear why a user would choose between Figure of Merit and F1. Experienced members of the professional community have condemned Kappa because of Kappa's severe conceptual flaws [20,21]. The numerator of Kappa is twice the numerator of Phi, while their denominators differ, thus Kappa and Phi give similar but not identical information. The complexities of the mathematical expressions for Kappa and Phi render their interpretations challenging. Numerous additional summary metrics of agreement exist that derive from Hits, Misses, False Alarms, and Correct Rejections. Fielding and Bell [22] list 13 such summary metrics, all of which fail to reveal the total information in the confusion matrix, fail to distinguish between quantity & allocation differences and fail to assign differential costs to False Alarms & Misses. Researchers should use a metric if and only if the metric aligns firmly with a clearly articulated criterion that expresses the project's particular goals. Some goals might relate to none of the popular metrics. For example, stakeholders in an application to flood mapping might prefer a criterion that relates to how a threshold influences spatial patterns concerning whether water covers roads that influence accessibility to potentially stranded victims. None of the metrics in Table 3 measure that characteristic.

Regardless of the criterion to select an optimal threshold, scientists should report the threshold's sizes in the format of Figure 1. Misses, Hits, False Alarms, and Correct Rejections are four crucial and straightforward sizes that express information sufficient to compute numerous summary metrics. It is easier to interpret Misses, Hits, False Alarms, and Correct Rejections than to interpret most of the metrics that reduce those four informative sizes into one less informative unitless number. Nevertheless, summary metrics remain popular and misunderstood, so future research should illuminate the behavior of such metrics over the domain of the TOC parallelogram.

### 4.3. Sample Design

The sample design requires two steps, both of which require some prior information. First is the creation of the strata then second is the allocation of the number of observations to each stratum. Effective creation of strata would have a high concentration of reference presence in some strata, an intermediate concentration in other strata, and a low concentration in the remaining strata. This would allow an allocation of observations to the strata that have an intermediate concentration of suspected reference presence, which is where uncertainty is greatest, thus where reference data are most informative. The number of sampled observations in a stratum and the size of the stratum determines the stratum's sampling intensity, which determines the weight of the observations in the stratum, which determines the distances between the points on the TOC curve. More sample observations generate shorter distances between the points and produce more detail on the TOC curve. Well-designed strata would allow scientists to concentrate observations where uncertainty exists or near important thresholds on the TOC curve.

The Strata baseline in Figure 4a shows that the union of Strata 1 and 2 is approximately the size of the Abundance, thus Stratum 3 is approximately the size of the Extent minus Abundance. This allows the Strata baseline TOC to have a threshold near the red star, which is an important threshold. Stratum 1 has the greatest sampling intensity, which proved to be an inefficient use of effort to collect reference data because the classification was certain in Stratum 1 where the reference data showed water's presence at all 50 observations. Fewer observations in Stratum 1 would have produced the same conclusion that water covers all of Stratum 1. The observations in Stratum 1 would have been more efficiently allocated to the other strata. The observations in Stratum 2 are 79% presence and in Stratum 3 are 80% absence, thus Strata 2 and 3 have nearly equal uncertainty. However, Stratum 2 has twice as many observations as Stratum 3 while Stratum 2 is smaller than Stratum 3, thus Stratum 2 has a greater sampling intensity than Stratum 3. The observations in Stratum 2 would have been more efficiently allocated to Stratum 3. These insights concerning sample design are possible after we collected and analyzed the reference data for this application. Such insights can inform sample designs for future applications where investigators must design the sample prior to the collection of the reference data.

*4.4. Software*

The first author of our manuscript used Python to create the TOC Curve Generator [23]. The TOC Curve Generator is an executable file for Windows, thus computers do not need any additional software. Users do not need to know any programming languages.

Figure 5 shows the interface and some output of the TOC Curve Generator. Figure 5a shows the interface to generate TOC curves from files with various extensions, such as TIFF, RST, XLSX, and TXT. Figure 5b shows the interface for customizing the TOC curves. Figure 5c is a plot with multiple TOC curves.

The software allows users to select the thresholds, rank a categorical variable, set weights for the strata, plot multiple TOC curves in one parallelogram, and use reference data from a census, simple random sample, or stratified random sample. The TOC package in the software R lacks some of these features [24]. Also, the TOC Curve Generator has a detailed instruction manual.

Users can select the sequence and values of thresholds in the group box named "Options" in Figure 5a. The selection of thresholds determines how much information the TOC curve shows. By default, the software computes the maximum number of thresholds by assigning a rank to every unique index value. However, if the number of unique index values is large, then the software's default might cause a burdensome computing duration. The software allows an option to create a customized list of fewer thresholds by setting the minimum threshold value, the maximum threshold value, and a constant increment between each consecutive threshold value. This option can reduce the run time.

The software has an option to rank a categorical variable according to each category's ratio of the number of reference presence observations to the number of observations. Therefore, the user can use a categorical variable without having a preconceived notion concerning the ranking of the categories. If two categories have the same ratio, then the user can assign an earlier rank to the category that has a higher suspicion based on the user's preconceived notions. In Figure 5a, users can switch to "Categories "to start using a categorical variable for TOC curves.

When users get reference data from a sample, they can set weights. The default weight for each observation is 1. Users can change the size of each stratum to change the weight. The software calculates the weight of each observation automatically and applies the weights when drawing the TOC curves.

The TOC Curve generator allows users to customize the TOC curves. The software allows a plot that has multiple TOC curves. Users can edit axis titles, index names, threshold markers, and threshold labels. Users have the option to add a uniform baseline and red stars where Diagnosed Presence equals Abundance. Users can save the output in multiple image formats for easy publication.

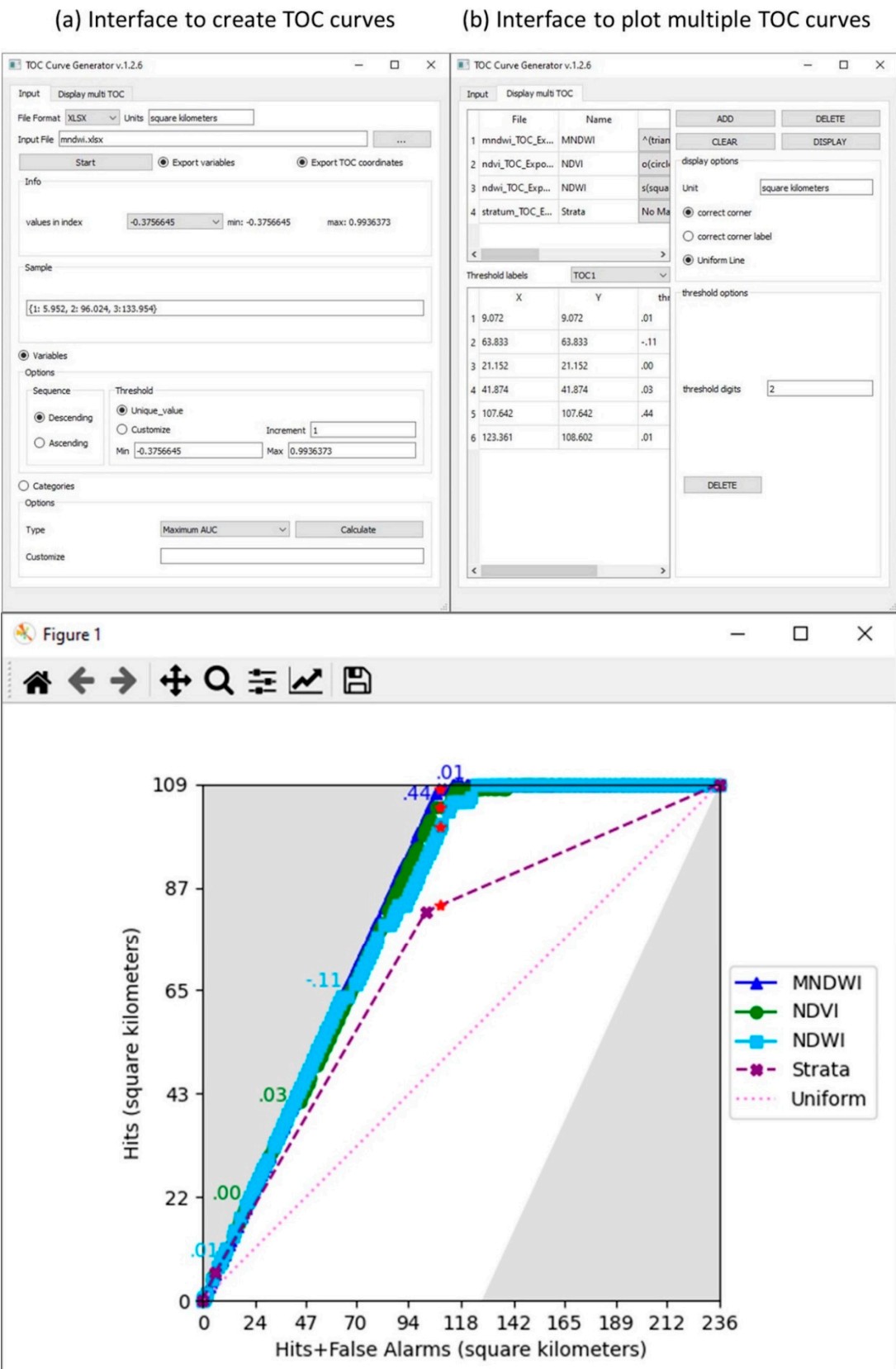

**Figure 5.** The TOC Curve Generator interface (**a**) the interface to specify inputs, (**b**) the interface to select display parameters, and (**c**) an example plot with multiple TOC curves.

*4.5. Limitations and Next Steps*

Future versions of The TOC Curve Generator will add capabilities that address some present limitations. The present version does not address variation due to sampling. The randomness of the sampling can cause each sample to generate variation in the resulting TOC curve. Smaller sample sizes are likely to cause more uncertainty. More uncertainty exists in longer segments of the TOC curve, especially when the segments have slopes parallel to the uniform line. Future versions will compute upper and lower bounds on the TOC curve and its AUC. Pontius Jr and Parmentier [25] offer a method to compute upper and lower bounds on the AUC for simple random sampling. However, the range between the upper and lower bounds might be smaller for stratified random sampling than for simple random sampling, depending on the number of pixels from each stratum in each segment of the TOC curve.

Section 4.2 explains why is it important to consider the cost of a False Alarm relative to the cost of a Miss. Future versions of the TOC Curve Generator will allow users to enter the relative costs, then label each threshold with the combined cost. Future versions will also allow users to label each threshold with a variety of metrics in Section 4.2.

Section 4.2 lists eight of the many metrics available to summarize a contingency table of the form in Figure 1. Future research should investigate the behavior of each of the metrics over the domain of the TOC parallelogram. Each metric might reach its optimum at a distinct quantity of diagnosed presence, in which case the optimum threshold depends on the data and the selection of the metric.

Our experience shows that the publication of concepts and the creation of free software are necessary but not sufficient for professionals to adopt new techniques that are better than traditional methods. Adoption by professionals requires video tutorials, conference presentations, and professional workshops. Our experience shows that scientists are reluctant to abandon traditional metrics, even when convincing literature exists. An example is that remote sensing professionals continue to use kappa, even when highly cited literature condemns kappa [20,21]. Courage and scientific integrity will be necessary for users to adopt the TOC and to abandon the ROC. Users will find the TOC more useful than the ROC because the TOC shows at each threshold the total information to reveal all the entries in the confusion matrix. The ROC shows at each threshold only two bits of information, which are insufficient to reveal the confusion matrix. Therefore, the TOC is more interpretable than the ROC.

## 5. Conclusions

Our manuscript's new methodological contribution shows how to use stratified random sampling to produce the Total Operating Characteristic. Previous methods required either simple random sampling or a census. A practical application to flood mapping shows that MNDWI has a greater ability than NDVI and NDWI to diagnose water for the illustrative case study. Furthermore, NDVI is superior to NDWI at most threshold values for the case study.

Our manuscript introduces free software to compute the Total Operating Characteristic. We encourage researchers to apply the TOC to compare the diagnostic characteristics of various indices. TOC curves are more informative and easier to interpret than ROC curves. TOC's concepts apply to a variety of applications in Remote Sensing and other professions. The TOC Curve Generator is the first, and so far only, software that can compare multiple TOC curves based on data that derive from a stratified random sample. The software works also for a simple random sample or census. The software is free at https://lazygis.github.io/projects/TOCCurveGenerator.

**Author Contributions:** Z.L. was responsible for data curation, formal analysis, software, and writing the original draft. R.G.P.J. was responsible for conceptualization, funding acquisition, supervision, and editing the draft. Both authors have read and agreed to the published version of the manuscript.

**Funding:** The Edna Bailey Sussman Fund supported this research via grant 24608. The United States National Science Foundation supported this research through the Long Term Ecological Research network to its Plum Island Ecosystems site via grant OCE-1637630. Sponsors played no role in data collection, analysis, interpretation, writing, or submission.

**Acknowledgments:** The authors thank Cloud to Street for the data.

**Conflicts of Interest:** The authors declare no conflict of interest. The founding sponsors had no role in the design of the study; in the collection, analyses, or interpretation of data; in the writing of the manuscript, and in the decision to publish the results.

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
