# Peer review of "The Total Operating Characteristic from Stratified Random Sampling with an Application to Flood Mapping"

_remotesensing, doi:10.3390/rs13193922_

Round 1

Reviewer 1 Report

Manuscript Ref. No.: remotesensing-1343614 Manuscript Title: “Stratified random sampling generates the Total Operating Characteristic with an application to flood mapping” Overview: The manuscript presents a new methodology to combine stratified random sampling with the total operating characteristic to compare various indices such as NDVI, NDWI, MNDWI. The methodology is applied to a hypothetical example and a practical application to flood mapping. However, the necessity to develop this approach is not highlighted by the authors. Also, the manuscript does not highlight the robustness of the approach through extensive statistical comparison. The software which is developed needs to be described in detail for its better understanding. The conclusions are loosely written and they do not highlighting the successful applicability of the new approach with its limitations and future scope of work. Overall, I recommend MAJOR REVISION of the manuscript. The authors can incorporate all the suggestions and provide necessary clarifications in the revised manuscript. Technical Comments: 1. The title needs modification. The word ‘generates’ seems to be improper. 2. The research gap is not properly highlighted in the Introduction part. Please identify the research gap suitably by pointing out the shortcomings of the previous studies. 3. The organization of the manuscript shall be included at the end of the Introduction. 4. Section 2.2: When the practical example is discussed, the case study region details should be described properly. Also, the indices such as NDVI, NDWI, MNDWI should be defined for benefit of the readers. 5. Page 11 Line 365: Of all the threshold selection criteria presented in Table 3, the authors recommend considering ‘quantity difference’ criteria. What is the justification or basis of it compared to other criteria? 6. Page 11 Lines 377-378: ‘However, the cost of a False Alarm differs from the cost of a Miss for many applications’. Justify the statement. 7. It would be more appropriate if the authors can present a tabular comparison of numerical values of all threshold selection criteria listed in Table 3 for the discussed practical problem. 8. Page 13 Line 450: The authors have cited Wikipedia reference, but it is not a reliable source. The authors are recommended to cite a published article for more reliability. 9. The description about the software is insufficient. The authors should provide some insights and features about the software probably by including a couple of figures about the software. Also, the authors should describe about the system requirements for the software and its source code language. 10. The authors must highlight the limitations of the study as well while using the new approach proposed by them. 11. The conclusions are very brief. They do not highlight all the key features of the developed methodology, its application and future scope of work regarding the same. Minor Editorial Comments: 1. Page 1 Line 40: Many researchers use MNDWI for flood mapping. Pls cite those studies? 2. Introduction: The word ‘researchers’ is used very often. Consider it replacing with ‘studies’ or some suitable word. 3. Figure 2: The notations H, F, C, M are not defined.

Reviewer 2 Report

The paper is interesting with practical implications. However, some minor revisions would be suitable:

- Introduction - Provide more discussions on practical applications of TOC in various fields of study, refering to relevant literature.

- Add aim of the paper at the end of the Introduction section. You may re-formulate the last paragraph to be the aim.

- 2.2 Practical Example - Add more description about study area and map of the study area.

- Lines 449-450 - Please, do not work with Wikipedia in scientific articles.

- Extend the Conclusion section with one or two paragraphs. Include also future research directions here.

Reviewer 3 Report

The manuscript is devoted to new methodology to generate Total Operating Characteristics applicable for flood mapping. The authors propose such generation with the use of stratified random sampling.

They demonstrate the efficiency of the method they propose with the practical example. Their method looks more informative than Relative Operating Characteristics, often used. The authors are so generous that they propose their software free for any researchers interested.

The paper looks rather interesting and may be printed.

Round 2

Reviewer 1 Report

The authors have significantly improved the manuscript. It can be accepted for publication.